# Effect of Integrin Blockade on Experimental Spondyloarthritis

**DOI:** 10.3390/biom14111386

**Published:** 2024-10-31

**Authors:** Enoch Yau, Melissa Lim, Zoya Qaiyum, Shaghayegh Foroozan Boroojeni, Michael Tang, Addison Pacheco, Fataneh Tavasolian, Robert D. Inman

**Affiliations:** 1Schroeder Arthritis Institute, Toronto Western Hospital, University Health Network, Toronto, ON M5G 2C4, Canada; enoch.yau@alumni.utoronto.ca (E.Y.);; 2Departments of Medicine and Immunology, University of Toronto, Toronto, ON M5S 1A8, Canada

**Keywords:** integrins, spondyloarthritis, SKG, biologics

## Abstract

Spondyloarthritis (SpA) describes a group of diseases characterized by chronic inflammation in the spine and peripheral joints. While pathogenesis is still unclear, proinflammatory gut-derived immune cells have been identified in the joints of SpA patients. We previously identified an enriched population of integrin-expressing cells in the joints of SpA patients. Entry of gut-derived cells into joints may be mediated by these integrins. In the current study, we used the SKG murine model of SpA to study the impact of integrin blockade. Mice were injected with antibodies against the integrin α4β7 or the β7 monomer twice a week. Treatment with antibodies against α4β7 reduced disease severity in curdlan-injected SKG mice, with disease scores being comparable between treatment initiation times. Targeting the β7 monomer led to reduced arthritis severity compared to targeting the α4β7 dimer. Treatment with antibodies against α4β7 or β7 decreased expression of these integrins in CD4+ T cells, with the frequency of αE+β7+ T cells in the spleen and lymph nodes correlating with disease severity. In summary, we showed that integrin blockade showed potential for ameliorating disease in a murine model of SpA, lending support for further studies testing integrin blockade in SpA.

## 1. Introduction

Spondyloarthritis (SpA) describes a group of diseases characterized by chronic inflammation in the spine and joints [1]. While the exact pathogenesis of SpA is still unknown, the gut has been proposed to play a role. Inflammatory bowel disease (IBD) has been reported to occur in up to 10% of axial SpA (axSpA) patients, and up to 60% of patients show signs of gut inflammation below the threshold for a diagnosis of IBD [2]. In addition, genome wide association studies have identified several shared genetic risk factors between SpA and IBD [3,4]. Several hypotheses have been proposed as to how gut and joint inflammation may be linked, including the aberrant trafficking hypothesis. This hypothesis posits that following activation in the gut, T cells and other immune cells can reenter circulation and home to joints, where they can drive inflammation [2]. Indeed, adoptive transfer and cell tracking experiments in SpA mouse models have demonstrated that gut-derived T cells could be found in inflamed joints of arthritic mice. Further, these gut-derived T cells demonstrated increased production of pro-inflammatory cytokines and arthritogenicity compared to non-gut-derived T cells [5,6].

Entry of these cells into joints is mediated by interactions between cell adhesion molecules expressed on T cells, such as integrins, and ligands expressed on joint endothelial cells [7]. The integrin α4β7 has been shown to mediate entry of immune cells into the gut through binding MAdCAM-1, a cell adhesion molecule expressed on gut endothelium [7]. Once in gut tissue, the integrin αEβ7 facilitates retention of immune cells through binding to the ligand E-cadherin, expressed by intestinal epithelial cells [7]. Evidence suggests that T cells may employ these integrins to similarly home to joints. MAdCAM-1 expression has been found on endothelial cells derived from human joints, and T cell lines derived from synovial biopsies of SpA patients demonstrate expression of α4β7 and αEβ7 [8,9,10]. Further, single cell profiling has identified a population of T cells expressing numerous integrin chains, including β7, that are enriched in the synovial fluid of SpA patients compared to blood [11].

Monoclonal antibodies targeting either the α4 chain or the α4β7 integrin dimer have demonstrated clinical benefit in treating IBD [12]. However, to date, evidence supporting integrin-targeting therapies for SpA is limited. At present, only one previously published study has tested the effect of integrin blockade on an animal model of SpA, though this focused on the signaling rather than the cell adhesion function of integrins [13]. In IBD patients, the effect of integrin blockade on articular disease is unclear. Some case reports have described new onset of SpA or flares of underlying SpA in patients following integrin blockade, while others have reported reductions in both IBD and SpA disease activity following integrin blockade [14,15,16,17,18,19]. Thus, we sought to determine the effect of blocking integrins on SpA using the SKG mouse model.

SKG mice feature a point mutation to the *Zap70* gene and develop an SpA-like disease, including chronic enthesitis, arthritis, and ileitis, following injection with the fungal cell wall derivative curdlan [20]. The role of T cells in disease pathogenesis has been well characterized in this model, and joint homing of gut-derived T cells has previously been demonstrated in SKG mice, making the curdlan-induced SKG mouse model suitable for assessing whether blocking integrins can affect SpA-like disease [5].

In this study, we found that treatment with monoclonal antibodies against the α4β7 integrin reduced disease severity in curdlan injected SKG mice, with greater benefit observed in mice treated with antibodies against the β7 monomer. Interestingly, disease scores were comparable between different treatment initiation times.

## 2. Materials and Methods

### 2.1. Mice, Disease Induction and Anti-Integrin Treatments

SKG mice were bred under SPF conditions. Disease was induced by injecting SKG mice aged 6–10 weeks intraperitoneally (i.p.) with 3mg of curdlan. Only female SKG mice were used due to a higher disease severity compared to male mice [20]. Following disease induction, animals were weighed and assessed for clinical signs of disease twice per week for 6 weeks post-curdlan. Starting from either 1 week before curdlan, 3 days post-curdlan, or 2 weeks post-curdlan, mice were injected i.p. twice per week on alternating sides with 200 μL rat anti-mouse integrin α4β7 antibody (BioXCell, BE0034, Lebanon, NH, USA), rat anti-mouse integrin β7 (BioXCell, BE0062), rat IgG2a isotype control (BioXCell, BE0089) or PBS. Antibody solutions were prepared fresh prior to each injection by diluting stock antibody solutions to 100 μg/dose or 400 μg/dose in PBS. Concentrations and dosing schedule were based on previous studies testing α4β7 blockade in murine models of disease [21,22,23]. All animal experiments were approved by the Animal Resource Centre of the University Health Network (UHN), Toronto, under the guidance of the Canadian Council on Animal Care and abide by the UHN’s policy on the Humane Use of Animal Models in Biomedical Research.

### 2.2. Clinical and Histological Scoring

Mice were weighed and assessed for clinical signs of disease by the same scorer twice per week. At each time point, an overall SpA score was calculated as well as dermatitis and arthritis scores for each mouse. At 6 weeks post-curdlan, mice were euthanized. Ankle, tail, and ileum samples were collected and processed into hematoxylin and eosin stained slides. Slides were histologically scored for signs of inflammation in a blinding fashion by two independent scorers and averaged.

### 2.3. Tissue Isolation

Upon sacrifice, spleen, popliteal and inguinal lymph nodes (PILNs), mesenteric lymph nodes (MLNs), Peyer’s Patches (PP), intraepithelial lymphocytes (IELs), and lamina propria lymphocytes (LPs) were collected from each mouse and processed into single cell suspensions following a previously described protocol, then stored in liquid nitrogen until flow cytometry analysis [24].

### 2.4. Flow Cytometry

Prior to antibody staining, cells were thawed and incubated with a viability dye (ThermoFisher, 65-0867-14, Waltham, MA, USA) and Fc block (Biolegend, 422302, San Diego, CA, USA). Cells were then stained with fluorescent antibodies against CD3 (Biolegend, 100204), CD4 (Biolegend, 100413), CD49a (Biolegend, 142606), CD49d (Biolegend, 103706), CD103 (Biolegend, 121417), CD69 (Biolegend, 104512), and integrin β7 (BD, 743791). Flow cytometry data were collected using the BD FACSCanto II flow cytometer and analyzed in FlowJo (version 10.6, Becton Dickinson, Franklin Lakes, NJ, USA).

### 2.5. Statistical Analysis

Statistical analysis comparing data between groups was performed by one-way, repeated measurements, two-way, or ordinary two-way analysis of variance (ANOVA) or Kruskal–Wallis tests where appropriate. Multiple comparisons were performed using Tukey’s or Dunn’s multiple comparisons tests as appropriate. For correlation analysis, Spearman coefficients were calculated. Two-tailed tests were used for all analyses, and *p*-values below 0.05 were considered statistically significant. All statistical analyses were performed using GraphPad Prism 9.

## 3. Results

### 3.1. SKG Mice Showed Dose Dependent Response to α4β7 Block

Treatment with α4β7 integrin blocking antibodies has demonstrated clinical benefit in IBD [12]. This is thought to be mediated by preventing integrin-mediated entry of inflammatory immune cells into the gut. Given that cells expressing multiple integrins, including α4β7, are enriched in the synovial fluid of SpA patients, it is possible that antibodies against α4β7 may provide similar benefit in treating SpA [9,11]. Therefore, we tested whether treatment with anti-α4β7 antibodies could ameliorate disease in the curdlan-driven SKG mouse model of SpA.

To mimic early clinical intervention of disease, we initiated integrin blockade treatment in SKG mice at two weeks post-curdlan, when clinical signs of disease typically begin to emerge [20]. As reported, SKG mice treated with curdlan developed SpA-like disease, including arthritis, dactylitis, and dermatitis. Mice that received the high but not the low dose of anti-α4β7 showed a significant reduction in mean SpA scores and an increase in weight compared to diseased controls (Figure 1A,B). While all diseased control mice developed arthritis and dermatitis by endpoint, only 3 of 5 high dose treated mice did. Consistent with this, high dose treated mice showed a significant decrease in arthritis severity and a trend towards decreased dermatitis severity compared to diseased controls (*p* = 0.12) (Figure 1C–F).

Following sacrifice, ankle, tail, and ileum samples from mice were collected and processed into H&E-stained slides to be scored for histological severity of disease. Mirroring what was seen clinically, ankle and tail slides derived from diseased control mice showed severe immune infiltration in the joint spaces, bones, tendons, and entheses (Figure 1G). Treatment with the high dose of the α4β7 blocking antibody significantly reduced histological signs of inflammation in the tail. While differences in histological signs of ankle disease between diseased controls and high dose treated mice did not reach statistical significance, two high dose treated mice showed no or minimal signs of histological ankle disease. These were the same mice that were clinically arthritis and dermatitis free at the experimental endpoint (Figure 1H,I). The remaining three high dose treated mice had comparable ankle disease scores to diseased controls, suggesting a potential split of responders and non-responders. Curiously, gut histological disease scores did not differ between any of the groups, with samples from all groups demonstrating some level of mucosal and submucosal immune infiltration (Appendix A and Figure 1J). This could possibly be driven by a basal level of gut inflammation, which has previously been reported in the colons of curdlan-naïve SKG mice [5].

### 3.2. Earlier Intervention with Integrin Blockade Was Not More Protective

Having observed clinical benefit in SKG mice treated with anti-α4β7 antibodies, we wanted to determine the effect of earlier initiation of integrin blockade. We hypothesized that earlier intervention could halt disease progression at an earlier stage and thereby provide additional benefit. We compared the effects of treatment with anti-α4β7 antibodies starting from one week pre-curdlan, 3 days post-curdlan, and 2 weeks post-curdlan.

As before, mice treated with the anti-α4β7 antibodies starting from 2 weeks post-curdlan showed significantly reduced SpA disease scores and improved weight gain compared to diseased controls at endpoint (Figure 2A,B). Unexpectedly, earlier intervention was not more protective than a 2 week start. In fact, mice that began treatment 1 week before curdlan or 3 days post-curdlan did not show statistically lower mean disease scores than diseased controls at endpoint (*p* = 0.72 and *p* = 0.07, respectively). Interestingly, regardless of when treatment began, clinical disease scores for integrin blockade-treated mice began deviating from those of diseased controls around two weeks post-curdlan.

Despite integrin blockade delaying the development of arthritis in treated mice, dermatitis did not seem to be affected by treatment in this experiment (Figure 2C,D). Treated mice showed reduced arthritis and dermatitis severity compared to diseased controls, but no clear trends with treatment initiation time could be seen (Figure 2E,F). Earlier initiation of integrin blockade treatment therefore did not appear to attenuate clinical signs of disease any more than a two week start point.

Of note, arthritis and dermatitis were not observed in mice that were pretreated with the integrin blockade, with arthritis onset in these mice still being delayed compared to diseased controls. Thus, treatment with an α4β7 inhibitor alone was not found to lead to a new onset of SpA in this experiment, as has been reported with human IBD patients [17,18].

Histologically, several mice treated with anti-α4β7 showed mild improvement in ankle disease scores compared to diseased controls, and this was true regardless of treatment initiation time, though mean differences did not reach statistical significance (Figure 2G). Similarly, a trend towards lower disease scores in the tails of treated mice compared to diseased controls could be seen, with the differences being the greatest in mice that began 2 weeks post-curdlan (*p* = 0.22) (Figure 2H). Gut histological disease scores showed no differences between treatment groups again (Figure 2I).

### 3.3. Blockade of the β7 Integrin Monomer Trended Towards Improved Disease Scores Compared to α4β7 Blockade

In addition to dimerizing with α4, the β7 integrin chain also dimerizes with αE in various immune cell types. Thus, blockade of β7 could theoretically inhibit gut inflammation through a dual effect on both integrins [25]. If these integrins are similarly involved in the entry and retention of pathogenic immune cells into joints, β7 blockade may provide therapeutic benefit in treating SpA. We hypothesized that treatment with antibodies against the β7 integrin chain would decrease signs of disease in the curdlan-induced SKG mouse model of SpA more than treatment with antibodies against the α4β7 dimer.

Indeed, we saw that anti-β7 treated mice showed a trend towards decreased disease scores compared to anti-α4β7 treated mice (*p* = 0.07) (Figure 3A), as well as improved weight gain compared to anti-α4β7 treated mice (*p* = 0.18) (Figure 3B). By endpoint, all diseased control mice developed arthritis, while only five of six treated with anti-α4β7 and four of six treated with anti-β7 did (Figure 3C). Anti-β7 treated mice had the lowest mean arthritis severity, with a significant reduction compared to anti-α4β7 treated mice. Dermatitis severity scores and incidence did not differ with treatment (Figure 3D–F). Thus, mice treated with anti-β7 showed reduced clinical signs of arthritis compared to anti-α4β7 treated mice.

Reflecting clinical findings, anti-β7 treated mice trended towards lower scores than anti-α4β7 treated mice in both the ankle (*p* = 0.30) and tail (*p* = 0.44) (Figure 3G,H). On the other hand, anti-β7 treatment did not seem to provide any benefit in signs of gut disease compared to anti-α4β7 (Figure 3I). In summary, SKG mice treated with anti-β7 trended towards improved clinical scores and histological scores in the ankles and tails compared to anti-α4β7 treated mice. Unexpectedly, no statistically significant differences were seen in clinical or histological disease scores between diseased controls and either anti-α4β7 treated mice or anti-β7 treated mice for this experiment.

### 3.4. Integrin Blockade Led to Tissue Specific Changes in Integrin Expression Among CD4+ T Cells

During inflammation, endothelial cells upregulate the expression of integrin ligands to facilitate immobilization of circulating immune cells [26,27]. Previous in vivo studies have demonstrated that integrin blockade led to redirection of immune cells from sites of inflammation to other tissues [28]. We therefore hypothesized that SKG mice treated with integrin blockade would demonstrate a reduction in integrin expression in sites of inflammation compared to diseased controls.

To test this, we isolated cells from the spleens, mesenteric lymph nodes (MLNs), and popliteal and inguinal lymph nodes (PILNs) of mice from the experiment shown in Figure 3 and quantified their integrin expression via flow cytometry. The MLN were representative of immune cells draining from the intestinal tissues, while the PILN were representative of immune cells draining from the ankles, hindfeet, and tail [24,29]. Since curdlan induced disease in SKG mice has been demonstrated to be driven by CD4+ T cells, we focused on integrin expression in CD3+CD4+ cells [20]. Gating strategy and representative plots are shown in Appendix A.

The α1 integrin chain (or CD49a) can dimerize with the β1 chain. α1β1 is expressed following T cell activation, particularly in inflamed tissues [30]. We therefore expected to see increased CD49a expression in CD4+ T cells of diseased SKG mice. Unexpectedly, no differences were seen in CD4+ T cell CD49a expression in the spleen and MLN between any groups. In the PILN, there was increased CD49a expression in the healthy compared to curdlan-treated mice, but no differences were seen between treated mice and diseased controls. Consistent with a role for α1β1 in tissue retention, expression was higher in the MLN and PILN compared to the spleen within the same treatment groups (Figure 4A).

Similarly, we expected to see an increase in surface expression of CD49d (or α4) in diseased SKG mice, but no significant differences were seen between healthy and diseased control mice in any of the analyzed tissues. However, as hypothesized, treatment with anti-α4β7 antibodies led to a significant reduction in CD49d expressing T cells in the MLN, and this was even greater for anti-β7 treated mice (Figure 4B).

CD103 (or αE) is only known to dimerize with the β7 chain. Consistent with the role of αEβ7 in retention in inflamed tissues, CD103 was significantly increased in the spleen and PILN and trended to an increase in MLN of diseased mice compared to healthy controls (*p* = 0.10) [31]. Treatment with anti-α4β7 led to reduced CD103 expression in all three analyzed tissues, while anti-β7 treatment drove even greater decreases (Figure 4C).

As with CD103, β7 expressing T cells were more frequent in diseased controls compared to healthy controls in all tissues analyzed. Anti-α4β7 treatment reduced β7 expression in all tissues, and anti-β7 treatment led to even greater reductions (Figure 4D). The anti-β7 antibody used for the in vivo blocking experiment (clone FIB504) targets a different epitope than the one used for flow cytometry analysis (clone M293), and these have previously been demonstrated to be non-competitive in binding to β7 [32].

Co-expression of CD69 and CD103 on T cells is traditionally used as a marker for tissue resident memory T cells. These cells rapidly produce inflammatory cytokines in peripheral tissues upon reactivation and have been implicated in flares of chronic immune-related diseases, including IBD and RA [33,34,35]. We found that the frequency of CD69+CD103+ T cells was increased in the spleens and MLN of diseased control mice compared to healthy controls but decreased with integrin blockade treatment, though the frequency of CD69+CD103+ T cells in the PILN was unexpectedly comparable between all groups (Figure 4E).

To see if expression of the analyzed surface markers was associated with disease severity, we correlated the frequencies of CD4+ T cells expressing these proteins in each tissue with the disease scores of corresponding mice. Generally, expression of these markers did not correlate with disease scores. However, expression of β7 in the PILN positively correlated with several measures of disease, including overall clinical scores, arthritis and dermatitis scores, and histological scores of the ankle and tail. This correlation may be indicative of a role for β7 in mediating immune cell homing and retention in the ankles and tails (Table 1). Curiously, a negative correlation between CD49d expression in the MLN and dermatitis scores was seen. This could represent a redirection of cells from the gut to the skin, as a recent study reported α4β7+CD4+ T cells that could home to both the gut and skin (Table 1) [36].

When we compared the frequency of α4+β7+ and αE+β7+ T cells with disease scores, we found that α4+β7+ T cell frequencies did not correlate with any disease scores. On the other hand, αE+β7+ expression in the spleen and PILN positively correlated with overall clinical scores, arthritis and dermatitis scores, and histological ankle and tail scores (Table 2). This may suggest that αEβ7 specifically is involved in mediating disease. The frequency of CD69+CD103+ T cells in the spleen and PILN also correlated with various measures of disease severity, suggesting a role for TRMs in the SKG mouse model (Table 3).

In summary, integrin blockade led to differential downregulation of various integrins on T cells in different lymphoid tissues. Expression levels of β7, CD69, and CD103 were correlated with disease severity, which may suggest that the attenuation of disease seen in integrin blockade-treated mice is driven through a mechanism involving integrin expressing T cells, such as prevention of integrin mediated entry and retention in inflamed tissues.

## 4. Discussion

Integrins have been implicated in the pathogenesis of a variety of diseases by facilitating immune trafficking into sites of inflammation [31]. Integrin expressing cells have been identified in the synovial fluid of SpA patients [11]. In animal models of SpA, immune cells have been demonstrated to traffic from the GI tract to joints, where they can subsequently produce inflammatory cytokines [5,6]. Whether these cells are entering joints by an integrin-mediated mechanism has yet to be determined.

To date, clinical reports of integrin blockade in SpA have been contradictory. A 2016 case report described significant improvement of inflammatory back pain in a patient with SpA and multiple sclerosis following treatment with natalizumab, an anti-α4 monoclonal antibody [14]. Conversely, a 2023 case report from our group described a patient with IBD who developed worsening back pain following treatment with vedolizumab, an anti-α4β7 monoclonal antibody, which improved when switched to an anti-TNF monoclonal antibody [19]. Clarifying the role of integrin-mediated trafficking in SpA could inform future therapeutic strategies, especially as integrin inhibitors have already been approved for treatment of IBD, a disease associated with SpA.

In this study, we demonstrated that mice treated with anti-α4β7 monoclonal antibodies showed a dose dependent improvement in clinical and histological disease scores compared to diseased controls. This suggests that integrins may play a pathogenic role in the SKG model of SpA, possibly through mediating cell entry into inflamed tissue, as has been shown in studies of IBD.

Surprisingly, we found that earlier initiation of treatment did not confer any additional benefit compared to mice that began treatment two weeks post-curdlan. Interestingly, regardless of when treatment began, the clinical scores of treated mice began diverging from diseased controls at around two weeks post-curdlan. This suggests that a key integrin-mediated event may be occurring around two weeks post-curdlan. Future experiments could determine if intervention during this timeframe is essential by testing whether mice that begin treatment later than two weeks post-curdlan still show attenuated disease.

Contrary to case reports that have described new onset of arthritis or flares of underlying arthritis in IBD patients following vedolizumab treatment, we did not report early onset or increased severity of arthritis in SKG mice that received anti-α4β7 antibodies before curdlan compared to controls [17,18,19]. One hypothesis regarding the association between vedolizumab treatment and arthritis posits that α4β7 blockade redirects immune cells from pre-existing sites of gut inflammation to the joints instead [19]. Treatment with oral DSS alone has been demonstrated to drive arthritis in SKG mice in an intestinal microbiome dependent manner [37]. Therefore, future experiments can test whether α4β7 blockade still ameliorates or instead worsens arthritis in SKG mice with more severe gut inflammation.

Since the integrin β7 chain can dimerize with both α4 and αE, we tested if targeting the integrin β7 chain as opposed to the α4β7 dimer would be more protective. Indeed, anti-β7 treated mice showed reduced arthritis severity compared to mice treated with anti-α4β7. Using flow cytometric analysis of lymphoid tissues in these mice, we demonstrated a gut-specific decrease in T cell α4 integrin expression and a broad decrease in T cell αE, β7, and CD69 expression in treated mice compared to diseased controls. Further, the expression of these integrins in the spleen and PILN of mice correlated with various disease scores, suggesting a role for αEβ7 and tissue resident memory T cells in the SKG disease model [38,39]. Monoclonal antibodies against the human β7 integrin chain are currently undergoing clinical trials for treatment of IBD [40]. The results presented here suggest targeting the β7 chain may provide additional therapeutic benefit for treatment of SpA.

During the experiment testing anti-β7, unexpectedly, anti-β7 and anti-α4β7 treated mice showed comparable clinical and histological disease scores as diseased controls, neither ameliorating nor exacerbating disease. Of note, however, disease scores in diseased control mice were lower in this experiment than previous ones. In addition, healthy controls unexpectedly showed persistent mild dermatitis throughout this experiment, and several mice in the experimental groups showed signs of dermatitis prior to curdlan administration. This has not been previously observed in studies using SKG mice in our lab. Inflammation prior to curdlan exposure may have explained why clinical disease scores for anti-α4β7 mice tracked closer with diseased controls than in the prior experiments.

While our experiments did not explicitly study immune cell trafficking, given the decrease in integrin expressing T cells in the MLN and PILN and previous work with colitis models, we propose a model whereby arthritogenic T cells in the SKG mouse model are entering joints from circulation through the use of the integrins α4β7 and αEβ7. Treatment with anti-integrin antibodies can therefore ameliorate disease in SKG mice through preventing integrin–ligand binding, excluding these T cells from joints, and preventing arthritis (Figure 5).

Although integrin blockade decreased mean disease scores of treated mice compared to diseased controls, this was not fully curative. The monoclonal antibody dosages for this study were chosen using dosages previously used in murine colitis models; future studies may therefore test if greater doses provide additional benefit to these mice. Assuming sufficient dosage, treated mice may have still shown signs of disease due to the exclusion of Tregs from inflamed tissue. In the TNF^ΔARE^ model of SpA, intestinal Tregs demonstrate the ability to traffic to joints [6]. Studies using colitis mouse models have found that treatment with anti-β7 antibodies exacerbated colitis due to the exclusion of Tregs from the gut [41,42]. Finally, disease in treated mice could be explained by pathogenic cells exploiting alternative methods, including other integrins, for entering inflamed tissue following integrin blockade.

Across our experiments, we did not observe any significant differences in histological disease scores in the gut between treated mice and diseased controls, despite observing differences in clinical signs of disease. All gut samples were taken from the ilea of mice; sampling the colon or other sites along the small intestine may have revealed clearer differences. Alternatively, but more unlikely, integrin blockade may have a differential effect on gut and joint inflammation in the SKG mouse model.

While the current experiments demonstrated a role for integrin expressing cells in disease, the origin of these cells was not determined. To address this, adoptive transfer experiments or cell labeling experiments can be combined with integrin targeting treatment to determine where integrin expressing cell traffic originates from and where they divert to following integrin blockade. Additional experiments may aim to investigate the effect of blockade of other integrins or immune associated cell adhesion molecules in an SpA model.

## 5. Conclusions

Using the curdlan-treated SKG mouse model, we demonstrated that integrin blockade could ameliorate SpA-like disease. Curdlan-treated mice showed a dose dependent response to monoclonal antibodies against the integrin α4β7, including decreased overall clinical scores, arthritis severity, arthritis and dermatitis incidence, and histological disease scores in the tail, with improved weight change compared to diseased controls. Protection from disease conferred by integrin blockade was not greater if treatment was initiated earlier, including prophylactic treatment. However, prophylactic treatment did not worsen disease as previously described in case reports of patients with IBD starting vedolizumab. Targeting the β7 integrin led to lower arthritis severity at endpoint compared to targeting the α4β7 dimer, likely due to a dual effect on α4β7 and αEβ7. Unexpectedly, however, in that experiment neither treatment led to significant worsening or improvement of clinical disease compared to diseased controls. Despite this, treatment with either antibody was shown to reduce expression of various integrins on T cells in the spleen and lymph nodes of mice, with greater effects seen in mice treated with anti-β7. T cell αE and β7 co-expression in the spleen and PILN correlated with disease scores across all treatment groups, suggesting a role for αEβ7 expression in disease in the SKG mouse model. Future experiments can aim to address the role of other immune trafficking mechanisms or the origin of pathogenic integrin-expressing cells.

The results of these experiments implicate a pathogenic role for integrins in the SKG mouse model of SpA and contribute towards a better understanding of the pathogenesis of SpA. Given that integrin inhibitors have already demonstrated efficacy in treatment of IBD, a disease closely associated with SpA, these findings have therapeutic implications and lend support for further studies testing integrin blockade in SpA.

## Figures and Tables

**Figure 1 biomolecules-14-01386-f001:**
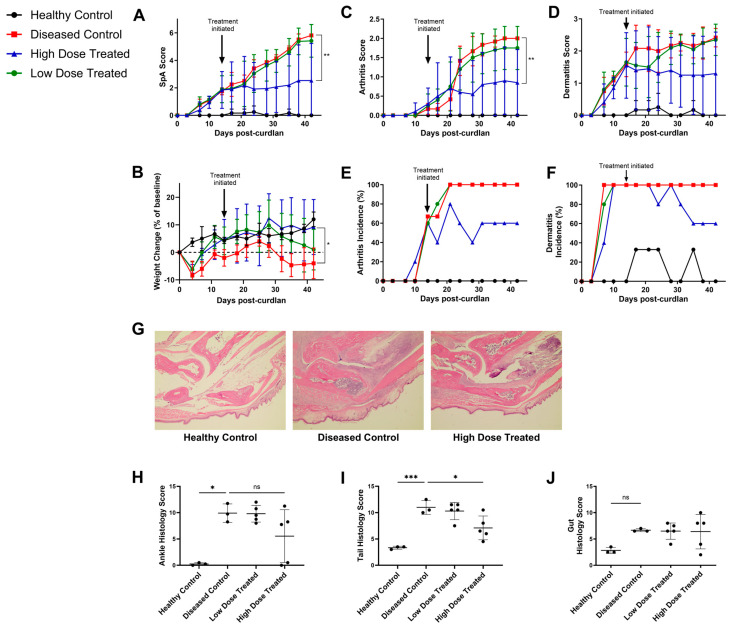
Improved signs of disease in SKG mice following α4β7 block. Female SKG mice aged 6–10 weeks were injected with PBS (Healthy Control) or curdlan and treated twice a week with rat anti-α4β7 antibodies at a high dose (400 μg/dose) or low dose (100 μg/dose) or with rat IgG2a isotype control antibodies (Diseased Control). (**A**) Clinical disease scores, (**B**) weight loss, (**C**) arthritis severity, (**D**) dermatitis severity, (**E**) arthritis incidence, and (**F**) dermatitis incidence of mice over 6 weeks after curdlan injection. (**G**) Representative hematoxylin and eosin-stained sections of ankles at 4× magnification. Histological disease scores for H&E-stained (**H**) ankle, (**I**) tail, and (**J**) gut slides. Data shown are means ± standard deviation. Statistical analysis was performed for (**A**–**D**) using a repeated measurements two-way ANOVA and Tukey’s multiple comparisons test and for (**H**–**J**) using a one-way ANOVA with Tukey’s multiple comparisons tests or Kruskal–Wallis test with Dunn’s multiple comparisons tests where appropriate. For control groups, n = 3, and for treated groups, n = 5. *: *p* < 0.05, **: *p* < 0.01, ***: *p* < 0.001, ns: not significant.

**Figure 2 biomolecules-14-01386-f002:**
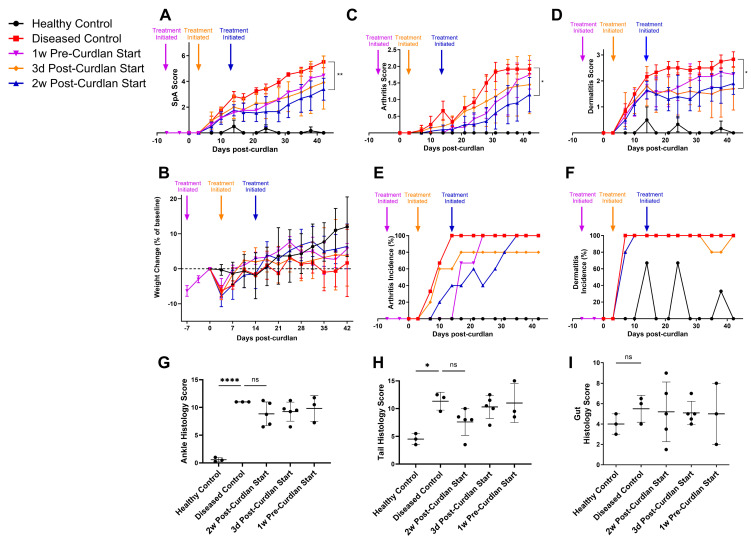
Comparable signs of disease in SKG mice with earlier initiation of integrin blockade treatment. Female SKG mice aged 6–10 weeks were injected with PBS (Healthy Control) or curdlan and treated twice a week with rat anti-α4β7 antibodies starting from 1 week pre-curdlan, 3 days post-curdlan, or 2 weeks post-curdlan, or with rat IgG2a isotype control antibodies starting 2 weeks post-curdlan (Diseased Control). (**A**) Clinical disease scores, (**B**) weight loss, (**C**) arthritis severity, (**D**) dermatitis severity, (**E**) arthritis incidence, and (**F**) dermatitis incidence of mice up to 6 weeks after curdlan injection. Histological disease scores for H&E-stained (**G**) ankle, (**H**) tail, and (**I**) gut slides. Data shown are means ± standard deviation. Statistical analysis was performed for (**A**–**D**) using a repeated measurements two-way ANOVA and Tukey’s multiple comparisons test and for (**G**–**I**) using a one-way ANOVA with Tukey’s multiple comparisons tests. For control groups and for 1 week pre-curdlan group, n = 3, and for 3 day and 2 weeks post-curdlan groups, n = 5. *: *p* < 0.05, **: *p* < 0.01, ****: *p* < 0.0001, ns: not significant.

**Figure 3 biomolecules-14-01386-f003:**
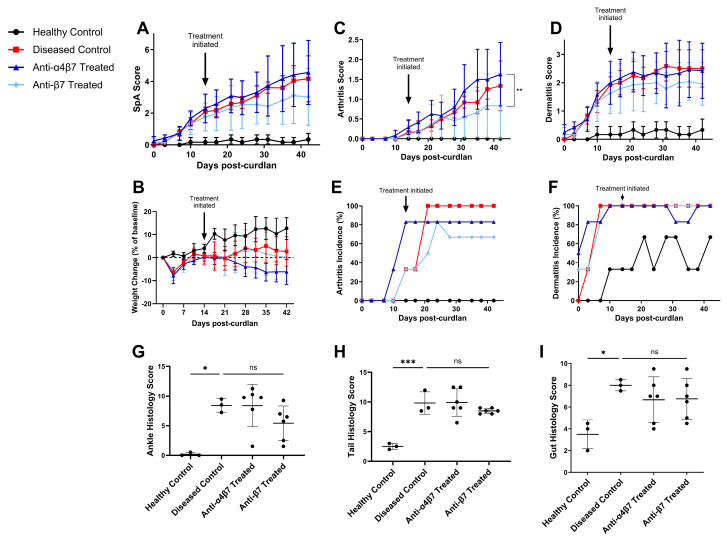
Mice treated with antibodies against the β7 integrin chain showed improved signs of disease compared to targeting α4β7. Female SKG mice aged 6–10 weeks were injected with PBS (Healthy Control) or curdlan and treated twice a week with rat anti-α4β7 or anti-β7 antibodies starting from 2 weeks post-curdlan, or with PBS (Diseased Control). (**A**) Clinical disease scores, (**B**) weight loss, (**C**) arthritis severity, (**D**) dermatitis severity, (**E**) arthritis incidence, and (**F**) dermatitis incidence of mice up to 6 weeks after curdlan injection. Histological disease scores for H&E-stained (**G**) ankle, (**H**) tail, and (**I**) gut slides. Data shown are means ± standard deviation. Statistical analysis was performed for (**A**–**D**) using a repeated measurements two-way ANOVA and Tukey’s multiple comparisons test and for (**G**–**I**) using a one-way ANOVA with Tukey’s multiple comparisons tests. For control groups, n = 3, and for treated groups, n = 6. *: *p* < 0.05, **: *p* < 0.01, ***: *p* < 0.001, ns: not significant.

**Figure 4 biomolecules-14-01386-f004:**
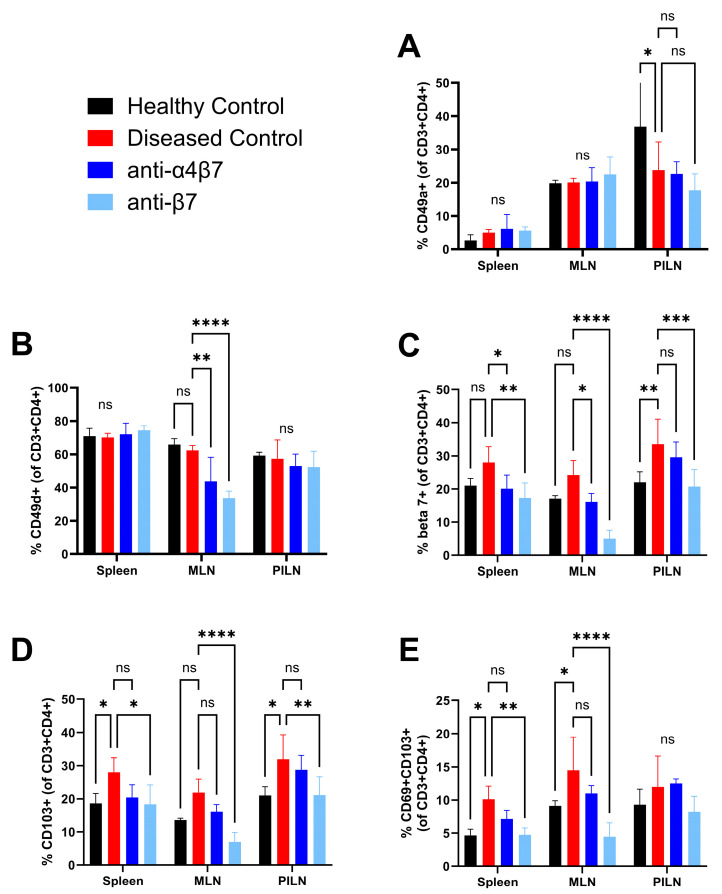
Integrin blockade drove tissue specific changes in T cell integrin chain expression. The frequency of CD3+CD4+ T cells expressing the integrin chains (**A**) CD49a, (**B**) CD49d, (**C**) CD103, (**D**) β7, or (**E**) co-expressing CD69 and CD103 from frozen spleen, mesenteric lymph node (MLN), or popliteal and inguinal lymph node (PILN) suspensions. For control groups, n = 3, and for treated groups, n = 6. Data shown are means ± standard deviation. Statistical analysis of histological scores was performed using an ordinary two-way ANOVA and Tukey’s multiple comparisons test. *: *p* < 0.05, **: *p* < 0.01, ***: *p* < 0.001, ****: *p* < 0.0001, ns: non-significant.

**Figure 5 biomolecules-14-01386-f005:**
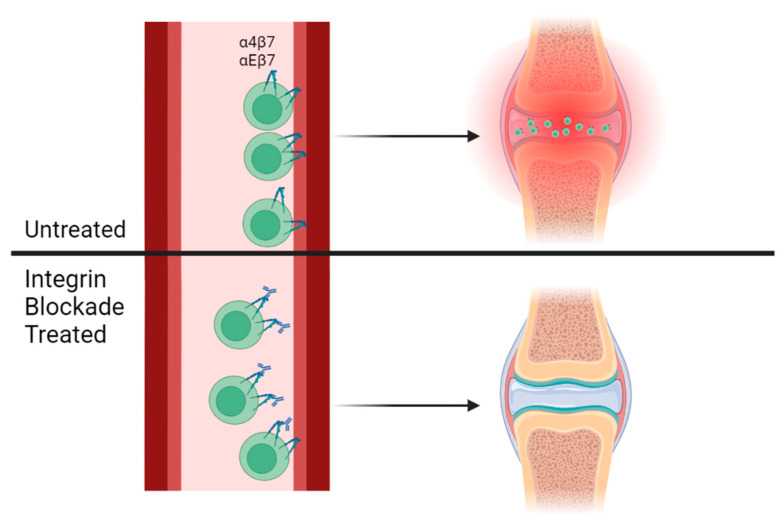
Integrin blockade may attenuate disease by preventing joint entry of arthritogenic T cells. In untreated mice, inflammatory T cells enter and remain in joints from circulation through the use of integrins such as α4β7 and αEβ7. Treatment with antibodies against these integrins prevents integrin–ligand binding, excluding immune cells from target sites of inflammation, including joints.

**Table 1 biomolecules-14-01386-t001:** Integrin β7 expression in popliteal and inguinal lymph nodes positively correlated with disease severity. The frequencies of CD3+CD4+ T cells expressing the integrin chains CD49a, CD49d, or β7 in frozen spleen, mesenteric lymph node, or popliteal and inguinal lymph node suspensions were correlated with clinical and histological disease scores, n = 18. Data shown are Spearman correlation coefficients. Cells highlighted were statistically significant correlations, yellow: *p* < 0.05, orange: *p* < 0.01.

Disease Measure	Spleen CD49a+	MLN CD49a+	P/ILN CD49a+	Spleen CD49d+	MLN CD49d+	P/ILN CD49d+	Spleen Beta 7+	MLN Beta 7+	P/ILN Beta 7+
Clinical Scores	0.175	−0.134	−0.064	0.057	−0.388	0.127	0.328	−0.053	0.512
Arthritis Scores	0.206	−0.058	−0.009	0.062	−0.335	0.242	0.352	−0.010	0.548
Dermatitis Scores	0.164	−0.128	−0.084	−0.020	−0.470	0.012	0.345	−0.061	0.518
Gut Scores	0.411	−0.029	−0.045	0.086	0.017	−0.326	0.058	0.239	0.252
Ankle Scores	0.420	−0.139	0.080	0.058	−0.187	0.066	0.294	0.098	0.666
Tail Scores	0.247	−0.213	0.041	−0.069	−0.312	−0.118	0.291	0.016	0.587

**Table 2 biomolecules-14-01386-t002:** Frequency of T cells co-expressing αE and β7 in the spleen and popliteal and inguinal lymph nodes positively correlated with disease severity measures. The frequencies of CD3+CD4+ T cells co-expressing the integrin chains CD49d and β7 or αE (CD103) and β7 in frozen spleen, mesenteric lymph node, or popliteal and inguinal lymph node suspensions were correlated with clinical and histological disease scores, n = 18. Data shown are Spearman correlation coefficients. Cells highlighted were statistically significant correlations, yellow: *p* < 0.05, orange: *p* < 0.01.

Disease Measure	Spleen a4+b7+	MLN a4+b7+	P/ILN a4+b7+	Spleen aE+b7+	MLN aE+b7+	P/ILN aE+b7+
Clinical Scores	−0.285	−0.264	0.079	0.575	0.210	0.515
Arthritis Scores	−0.238	−0.204	0.204	0.540	0.224	0.538
Dermatitis Scores	−0.312	−0.290	−0.040	0.595	0.216	0.519
Gut Scores	−0.288	0.121	−0.182	0.176	0.383	0.338
Ankle Scores	−0.311	−0.172	0.146	0.498	0.377	0.690
Tail Scores	−0.446	−0.247	−0.032	0.562	0.319	0.650

**Table 3 biomolecules-14-01386-t003:** Frequency of T cells co-expressing αE and CD69 in the spleen and popliteal and inguinal lymph nodes positively correlated with disease severity measures. The frequencies of CD3+CD4+ T cells co-expressing CD69 and CD103 in frozen spleen, mesenteric lymph node, or popliteal and inguinal lymph node suspensions were correlated with clinical and histological disease scores, n = 18. Data shown are Spearman correlation coefficients. Cells highlighted were statistically significant correlations, yellow: *p* < 0.05, orange: *p* < 0.01.

Disease Measure	Spleen CD69+ CD103+	MLN CD69+ CD103+	P/ILN CD69+ CD103+
Clinical Scores	0.586	0.193	0.432
Arthritis Scores	0.614	0.228	0.447
Dermatitis Scores	0.531	0.184	0.387
Gut Scores	0.284	0.400	0.227
Ankle Scores	0.668	0.422	0.690
Tail Scores	0.594	0.324	0.530

## Data Availability

All necessary data are presented in this paper. Additional data may be available upon request.

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
