# Peer review of "Effect of Integrin Blockade on Experimental Spondyloarthritis"

_biomolecules, 2024, doi:10.3390/biom14111386_

Round 1
Reviewer 1 Report
Comments and Suggestions for Authors
The study by Yau et al. focused on the evaluation of specific integrin blockade (alpha4beta7 or beta 7) towards a potential effect on human SpA like disease development in SKG mice. Although this is a timely topic and of interest to the research community, there are some important points which need to be addressed.
A major concern is that multiple findings throughout the manuscript are described as “trends” rather than representing true biological effects, which might be (partly) due to the limited sample size for the specific experiments. In this regard, the authors could show p-values for indicated results in the text to understand the significance of the anti-integrin treatment responses. Moreover, the authors state “anti-β7 treated mice showed reductions in both clinical and histological signs of disease compared to mice treated with anti-α4β7” related to data shown in figure 3 (while no significant difference was observed between anti-B7 treated and disease control mice). This statement is disputable as – in contrast to the initial data shown in figure 1 and 2 – there was no significant effect between anti- α4β7 treatment and disease control mice (as also recognized by the authors in the discussion section, but not mentioned in the results). The authors should consider to repeat the experiment to (a) confirm the α4β7protective effect and (b) to make valid conclusions on a potential differential effect between anti-alpha4beta7 and beta7 blockade in this model.
Minor points:
-Not clear for which experiments the additional immune phenotyping analyses (flow data) were performed. In this regard, in figure 1H, 2 out of 5 mice showed complete protection on ankle histology suggesting responders and non-responders for the high dose α4β7 treatment. How is this related to clinical scoring in figure 1C and were immune parameters evaluated for these mice?
-Please describe the gut pathology phenotype in this model and show representative histological images (if needed in supplemental). Of note, anti-a4ß7 treated mice showed an increase in weight compared to disease controls although no changes were seen regarding gut histology (ileitis). Although not a hallmark of this model, did the author consider to evaluate large intestine features under these conditions? Authors allude on basal level of gut inflammation to be potentially present in curdlan-naïve SKG mice. Show representative images to support this.
-Regarding statistics for clinical scoring: was repeated measurements testing applied?
-Please show representative flow cytometry plots in relation to immune phenotyping.
Author Response
Comment 1: The study by Yau et al. focused on the evaluation of specific integrin blockade (alpha4beta7 or beta 7) towards a potential effect on human SpA like disease development in SKG mice. Although this is a timely topic and of interest to the research community, there are some important points which need to be addressed.
A major concern is that multiple findings throughout the manuscript are described as “trends” rather than representing true biological effects, which might be (partly) due to the limited sample size for the specific experiments. In this regard, the authors could show p-values for indicated results in the text to understand the significance of the anti-integrin treatment responses.
Response 1: We agree with the reviewer’s assessment that our limited sample size likely contributed to being unable to show statistically significant differences. P values were added in text where differences were described as trends for transparency.
Comment 2: Moreover, the authors state “anti-β7 treated mice showed reductions in both clinical and histological signs of disease compared to mice treated with anti-α4β7” related to data shown in figure 3 (while no significant difference was observed between anti-B7 treated and disease control mice). This statement is disputable as – in contrast to the initial data shown in figure 1 and 2 – there was no significant effect between anti- α4β7 treatment and disease control mice (as also recognized by the authors in the discussion section, but not mentioned in the results). The authors should consider to repeat the experiment to (a) confirm the α4β7protective effect and (b) to make valid conclusions on a potential differential effect between anti-alpha4beta7 and beta7 blockade in this model.
Response 2: Given the timeline to respond to reviewer comments, we are unable to repeat the experiment. The reviewer is correct in stating that 1) contrary to Figures 1 and 2, no significant difference was seen between anti-alpha4beta7 treated mice and diseased controls and 2) no significant effect was seen between the anti-beta7 treated mice and diseased controls. However, we believe the quoted statement remains valid as the comparison made was strictly between anti-beta7 treated mice and anti-alpha4beta7 treated mice. A statement was added at the end of results section 3.3 to recognize the lack of statistically significant differences between both the anti-beta7 and anti-alpha4beta7 treated mice to diseased controls. Changes were made to the abstract, discussion and conclusion sections to make more valid conclusions in light of this.
Comment 3: Not clear for which experiments the additional immune phenotyping analyses (flow data) were performed.
Response 3: Agree that wording was ambiguous for which experiment the flow data came from. Phrasing changed from “mice from anti-β7 experiment” to “mice from the experiment shown in Figure 3”
Comment 4: In this regard, in figure 1H, 2 out of 5 mice showed complete protection on ankle histology suggesting responders and non-responders for the high dose α4β7 treatment. How is this related to clinical scoring in figure 1C and were immune parameters evaluated for these mice?
Response 4: Agree that histological scores should have been correlated with clinical scores. The last paragraph of section 3.1 was edited to include this and to mention the potential for responders and non-responders. Unfortunately, lymphoid tissues were not collected from the mice in the experiment represented by Figure 1 and so immune profiling was not done.
Comment 5: Please describe the gut pathology phenotype in this model and show representative histological images (if needed in supplemental). Of note, anti-α4ß7 treated mice showed an increase in weight compared to disease controls although no changes were seen regarding gut histology (ileitis). Although not a hallmark of this model, did the author consider to evaluate large intestine features under these conditions? Authors allude on basal level of gut inflammation to be potentially present in curdlan-naïve SKG mice. Show representative images to support this.
Response 5: Representative images of ileal histological slides were included as Supplemental Figure S1. These illustrate a comparable gut pathology phenotype between mouse groups and demonstrate that curdlan naïve mice showed gut inflammation. However, colon samples were not collected during our experiments. We have added a section in the discussion on gut histology scores and included this as a limitation.
Comment 6: Regarding statistics for clinical scoring: was repeated measurements testing applied?
Response 6: Yes, statistical analysis of clinical scores used repeated measurements two-way ANOVA. Methods and figure captions were updated to clarify this.
Comment 7: Please show representative flow cytometry plots in relation to immune phenotyping.
Response 7: Gating strategy with representative flow cytometry plots added as Supplemental Figure S2.
Reviewer 2 Report
Comments and Suggestions for Authors
Dear Authors, thank you for this interesting manuscript. Nice and well conducted experiments. Using this mice model for SpA and the experimental use of antibodies against integrins components the SpA-like disease could be ameliorated in its expressions, (clinical scores, arthritis evaluations, histological scores). Given us and potential readers more information about de role if integrins in this kind of diseases. I really enjoyed your work and I do not have questions and concerns.Regards.
Suggestion:
1. If posible, but not essential could be if some acute phase reactants can be measured. Same for other molecules like IL-6, or TNFa could be evaluated.
2. Is there any other integrin involved in this immune process?
Author Response
Comment 1: Dear Authors, thank you for this interesting manuscript. Nice and well conducted experiments. Using this mice model for SpA and the experimental use of antibodies against integrins components the SpA-like disease could be ameliorated in its expressions, (clinical scores, arthritis evaluations, histological scores). Given us and potential readers more information about de role if integrins in this kind of diseases. I really enjoyed your work and I do not have questions and concerns. Regards.
If possible, but not essential could be if some acute phase reactants can be measured. Same for other molecules like IL-6, or TNFa could be evaluated. Is there any other integrin involved in this immune process?
Response 1: A LegendPlex assay for circulating cytokines in serum was performed but was not informative. Testing blockade of other integrins was added as a potential future direction in the discussion.
Round 2
Reviewer 1 Report
Comments and Suggestions for Authors
The authors have accurately addressed my questions.
Minor points regarding flow data is Figure S2:
- The L/D versus CD3 plot suggests the majority of T cells were non-viable upon staining. Was this a consistent finding? Please indicate whether these were representative plots from a healthy or diseased mouse, either treated or untreated.
- The data are clear but the resolution of the flow plots is not optimal so might be addressed if possible.
Author Response
Comment: The authors have accurately addressed my questions. Minor points regarding flow data is Figure S2:
-The L/D versus CD3 plot suggests the majority of T cells were non-viable upon staining. Was this a consistent finding? Please indicate whether these were representative plots from a healthy or diseased mouse, either treated or untreated.
-The data are clear but the resolution of the flow plots is not optimal so might be addressed if possible.
Response: Spleen and lymphoid tissue samples were frozen in liquid nitrogen upon harvest and analyzed via flow cytometry at a later date. The freezing and thawing process likely contributed to the large number of non-viable T cells which we consistently saw across different tissue samples and mice. Methods section was updated to mention that cells were frozen on harvest and later thawed for flow cytometry. Caption for Figure S2 has been updated to state the representative plot was from the MLN of a diseased control mouse (untreated). Higher resolution Figure S2 was provided